# SOX2 Promotes Invasion in Human Bladder Cancers through MMP2 Upregulation and FOXO1 Downregulation

**DOI:** 10.3390/ijms232012532

**Published:** 2022-10-19

**Authors:** Qipeng Xie, Xiaohui Hua, Chao Huang, Xin Liao, Zhongxian Tian, Jiheng Xu, Yunping Zhao, Guosong Jiang, Haishan Huang, Chuanshu Huang

**Affiliations:** 1Oujiang Laboratory (Zhejiang Lab for Regenerative Medicine, Vision and Brain Health), School of Laboratory Medicine and Life Sciences, Wenzhou Medical University, Wenzhou 325000, China; 2Department of Clinical Laboratory, The Second Affiliated Hospital & Yuying Children’s Hospital of Wenzhou Medical University, Wenzhou 325035, China; 3Department of Urology, Union Hospital, Tongji Medical College, Huazhong University of Science and Technology, Wuhan 430022, China

**Keywords:** SOX2, SKP2, FOXO1, nucleolin, bladder cancer invasion

## Abstract

SOX2, a member of the SRY-related HMG-box (SOX) family, is abnormally expressed in many tumors and associated with cancer stem cell-like properties. Previous reports have shown that SOX2 is a biomarker for cancer stem cells in human bladder cancer (BC), and our most recent study has indicated that the inhibition of SOX2 by anticancer compound ChlA-F attenuates human BC cell invasion. We now investigated the mechanisms through which SOX2 promotes the invasive ability of BC cells. Our studies revealed that SOX2 promoted SKP2 transcription and increased SKP2-accelerated Sp1 protein degradation. As Sp1 is a transcriptionally regulated gene, *HUR* transcription was thereby attenuated, and, in the absence of HUR, *FOXO1* mRNA was degraded fast, which promoted BC cell invasion. In addition, SOX2 promoted BC invasion through the upregulation of nucleolin transcription, which resulted in increased *MMP2* mRNA stability and expression. Collectively, our findings show that SOX2 promotes BC invasion through both SKP2-Sp1-HUR-FOXO1 and nucleolin-MMP2 dual axes.

## 1. Introduction

Bladder cancer (BC) is one of the most lethal diseases in developed countries, with an estimated 81,180 new incidences and 17,100 deaths in the United States in 2022 [1]. Worldwide, the estimated new incidence of BC is 573,278, and the total number of deaths is 212,536 [2]. Non-muscle-invasive bladder cancer (NMIBC) and muscle-invasive bladder cancer (MIBC) are two common types of human BC [3,4]. MIBC exhibits sarcomatoid and squamous features as well as highly aggressive behaviors such as increased invasion and metastasis [3,5]. MIBC often metastasizes to pelvic and iliac lymph nodes as well as visceral sites, such as the lungs, liver, and bone. At least 50% of MIBC patients die from metastasis within two years, and the 5-year survival rate of metastatic BC is only 6% [6]. Thus, the elucidation of the mechanisms of the invasion and metastasis of BC is of high significance to alleviate the mortality of MIBC.

The abnormal expression of SOX2 has been reported in many types of cancer, including lung cancer [7], breast cancer [8], colorectal cancer [9], and bladder cancer [10,11]. SOX2 is mainly considered as a molecular marker of cancer stem cells. Moreover, in many types of cancer, the overexpression of SOX2 correlates with tumor invasions, such as melanoma [12], ovarian cancer [13], hepatocellular carcinoma [14], and laryngeal cancer [15]. Our most recent studies indicate that the anticancer drug ChlA-F can antagonize BC invasion through the inhibition of SOX2 expression, suggesting an important role of SOX2 in mediating human BC invasion. However, the molecular mechanism underlying SOX2’s promotion of BC invasion has not been explored yet. In the current study, we investigated the role and mechanism of SOX2 in promoting the invasive ability of human BC cells. We found that SOX2 overexpression could initiate nucleolin transcription and expression, which in turn stabilized *MMP2* mRNA, consequently leading to BC invasion. We also found that SOX2 overexpression elevated SKP2 (S-phase kinase-associated protein 2) expression, which accelerated Sp1 protein degradation, thereby reducing Sp1-regulated FOXO1 expression and ultimately promoting BC invasion.

## 2. Results

### 2.1. SOX2 was Overexpressed in Human and Mouse BC Tissues and Contributed to BC Cell Invasion

Our previous studies demonstrate that the application of butyl-N-(4-hydroxybutyl)nitrosamine (BBN) in drinking water to mice for over 20 weeks results in 100% muscle-invasive BCs (MIBC) [16]. To study the potential involvement of SOX2 in BC development, mice were exposed to BBN (0.05%) in drinking water for 0, 9, 12, and 20 weeks (*n* = 15), and all bladder tissues were collected at the indicated time points. Upon the BBN-treated time increase, the bladder epithelial tissue gradually became malignant, and MIBC was developed at 20 weeks after BBN exposure. Immunohistochemistry (IHC) staining was then employed to examine the expression level of SOX2 in these mouse BC tissues (Figure 1A). The results indicated that SOX2 protein levels in the bladder tissues of mice treated with BBN for 20 weeks were significantly increased as compared with vehicle-control bladder tissues. (Figure 1B). Using the Kaplan–Meier Online Tool (http://kmplot.com/analysis/, accessed on 4 September 2022) [17], we found that BC patients with high expressions of SOX2 exhibited a decreased overall survival than patients with low expressions of SOX2 (Figure 1C). These results indicated that SOX2 expression might contribute to the progression and metastasis of human BC. Then, the SOX2 protein expression in the mouse bladder tissues was also examined by Western blot. The SOX2 protein expression in BBN-treated mouse bladder tissues was significantly higher than that in vehicle-control bladder tissues (Figure 1D, *n* = 5). The T24/T24T system is a mature BC model involving T24 cells with limited cancer metastasis and T24T cells derived from T24 cells with significant BC cell lung metastatic ability. Significant metastatic differences between the two cell lines provide an ideal model for studying the molecular basis of the invasion and metastasis of BC [18]. The SOX2 protein expression in T24T was markedly higher than that in T24 cells (Figure 1E). The above results indicated that SOX2 might be related to the invasion ability of BC. To determine whether SOX2 was required for BC cell invasion, SOX2 overexpressed plasmid was transfected into T24 cells and established T24(Vector) and T24(SOX2) stable cell lines as identified in Figure 1F. The invasion abilities of SOX2 overexpressed cells were significantly increased as compared to those of the vector transfectants (Figure 1H,I). In contrast, using CRISPR/Cas9 systems, we also knocked out the *SOX2* gene in T24T cells. The stable single-clone *SOX2* knockout transfectants T24T(KO*SOX2* C1) and T24T(KO*SOX2* C2) were selected for our further investigations (Figure 1G). *SOX2* knockout remarkably decreased T24T cell invasion abilities in comparison to its nonsense transfectants (Figure 1J,K). It was noted that the alteration of the SOX2 expression level specifically modulated BC invasion without affecting BC cell migration in a transwell assay (Figure 1H–K). Consistent with the results observed in the transwell assay, the wound healing assay also showed that the overexpression of SOX2 and the knockout of SOX2 did not affect BC cell migration (Appendix A). To validate our in vitro findings in vivo, we injected T24T(Vector) and T24T(KO*SOX2* C1) cells into the tail vein of nude mice and compared their ability to metastasize to the lungs in nude mice. The results indicated that the lung metastasis ability of the cells was significantly decreased after the knockout of SOX2 (Figure 1L,M). Consistently, the results obtained from analyses of the overall survival of MIBC patients using the Kaplan–Meier online tool (KM plotter) revealed that the high expression of SOX2 was associated with a shorter survival time of MIBC patients (Appendix A). These results demonstrated that SOX2 served as an essential oncogene responsible for promoting human BC cell invasion.

### 2.2. Upregulation of MMP2 and Downregulation of FOXO1 Play a Crucial Role in SOX2 Promoting the Invasive Ability of BC Cells

To understand the mechanism by which SOX2 regulates BC cell invasion, we compared essential proteins related to the regulation of BC invasion including RhoGDIα, RhoGDIβ, Src FOXO1, and MMP2 between T24 and T24T, T24(Vector) and T24(SOX2), and T24T(Vector) and T24T(KOSOX2 C1/C2) cells. As shown in Figure 2A–C, RhoGDIα, RhoGDIβ, and Src did not show consistent changes. Our previous studies have shown that FOXO1 and MMP2 are targeted by the anticancer compound ISO for its suppression of bladder cancer invasion [16]. MMP2 protein expression was profoundly increased in T24T cells in comparison to that in T24 cells. As expected, MMP2 was also markedly upregulated in T24(SOX2) but dramatically decreased in T24T(KO*SOX2*) cells compared to their corresponding scramble vector transfectants. Furthermore, we knocked down *MMP2* in T24(SOX2) cells and overexpressed MMP2 in T24T(KO*SOX2* C1) cells and found that the knockdown of MMP2 in T24(SOX2) cells impaired cell invasion due to SOX2 overexpression, while the ectopic expression of MMP2 rescued the invasive phenotypes observed in the SOX2 knockout T24T cells (Appendix A). To our interest, FOXO1 protein expression was opposite to that of MMP2 in Figure 2A–C as shown in the indicated cells. These results suggested that MMP2 and FOXO1 might be involved in SOX2-promoted BC cell invasion. To test whether FOXO1 contributed to the decrease in BC cell invasion, an ectopic FOXO1 with a Flag-tag construct was overexpressed in T24T cells (Figure 2D). The results showed that T24T(Flag-FOXO1) cells had decreased invasion ability when compared to T24T(Vector) cells in a transwell assay (Figure 2E,F), suggesting that FOXO1 might be a downstream effector of SOX2 and negatively regulated BC cell invasion abilities. At the same time, FOXO1 was knocked down in T24T(KO*SOX2*) cells and identified the effect as shown in Figure 2G. FOXO1 protein was dramatically reduced in T24T(KO*SOX2*/sh*FOXO1*#1&#2) cells compared to T24T(KO*SOX2*/Vector) cells. The transwell assay also showed that T24T(KO*SOX2*/sh*FOXO1*#2) cells had increased invasive ability as compared to T24T(KO*SOX2*/Vector) cells (Figure 2H,I). However, there was no change in the expression of the MMP2 protein (Figure 2G), suggesting that the inhibition of cell invasion by FOXO1 was not associated with MMP2. We further knocked down MMP2 in T24(SOX2) and the overexpression of MMP2 in T24T (KO*SOX2*) cells (Appendix A), and there was no significant change in the FOXO1 protein, further indicating that the FOXO1 and MMP2 pathways are independent axes from each other in mediating BC invasion due to SOX2 overexpression (Appendix A). These results revealed that SOX2 promotes the invasion of BC cells by upregulating MMP2 and downregulating FOXO1 pathways, respectively.

### 2.3. HUR Downregulation Mediated SOX2 Destabilization of FOXO1 mRNA

Given our results showing that SOX2 was important for the downregulation of *FOXO1*, our subsequent efforts were directed at identifying the mechanisms behind the SOX2-mediated downregulation of FOXO1. The *FOXO1* mRNA levels were firstly determined and they were dramatically decreased in T24T cells as compared with T24 cells (Figure 3A) but increased in T24T(KO*SOX2*) cells as compared with T24T(Vector) cells (Figure 3B). Then, the transcription activity of FOXO1 was evaluated by using a FOXO1 promoter-driven luciferase reporter. The result showed that the *FOXO1* promoter activity showed no observable change in T24 vs. T24T cells (Figure 3C), T24(Vector) vs. T24(SOX2), or T24T(Vector) vs. T24T(KO*SOX2*) (Appendix A), excluding the possibility of SOX2 inhibiting *FOXO1* transcription. We next evaluated whether there was any effect of SOX2 on *FOXO1* mRNA stability. The mRNA degradation assay showed that *FOXO1* mRNA in T24T cells was degraded faster than that in T24 cells (Figure 3D). These results suggested that SOX2 negatively regulated FOXO1 at the mRNA degradation level. The degradation of mRNA can be controlled by cis-acting sequence elements or trans-acting factors [19]. Some RNA-binding proteins, such as HUR, Nucleolin (NCL), and AUF1, have been reported to bind to their target mRNAs and regulate mRNA stability [20,21,22]. Therefore, we examined whether these RNA-binding proteins were involved in the regulation of *FOXO1* mRNA stability by SOX2. The downregulation of HUR protein expression was observed in T24T and T24(SOX2) cells as compared with their related scramble control transfectants (Figure 3E,F). Consistently, HUR was upregulated in T24T(KO*SOX2*) cells as compared with T24T(Vector) cells (Figure 3G). These results suggested that SOX2 might reduce *FOXO1* mRNA stability by downregulating HUR. To determine whether HUR was required for *FOXO1* mRNA stability, shRNAs specifically targeting human HUR (sh*HUR*#1 and sh*HUR*#2) were transfected into T24T (KO*SOX2*) cells and established T24T(KO*SOX2*/nonsense), T24T(KO*SOX2*/sh*HUR*#1), and T24T(KO*SOX2*/sh*HUR*#2) stable transfectants. As shown in Figure 3H, the HUR and FOXO1 protein levels were reduced in the T24T(KO*SOX2*/sh*HUR*) cell lines as compared with the T24T(KO*SOX2*/nonsense) cell lines. To detect whether HUR binds to *FOXO1* mRNA, we used an RNA-IP assay, in which an anti-GFP antibody was used to pull down all mRNAs that physically interact with the GFP-HUR protein. The mRNA was then extracted from the precipitated complex, and the presence of *FOXO1* mRNA was detected by RT-PCR. As shown in Figure 3I, *FOXO1* mRNA was found to be specific in the immune complex extracted from 293T(GFP-HUR) but not in 293T(GFP), strongly suggesting that HUR does interact with *FOXO1* mRNA to enhance its stability. The above results demonstrated that SOX2 downregulated FOXO1 expression through depleting HUR and in turn decreasing *FOXO1* mRNA stability.

### 2.4. SOX2 Inhibited HUR Transcription by Downregulating Sp1 Protein Expression

The *HUR* mRNA levels in T24T cells were dramatically lower than in T24 cells (Figure 4A). However, after SOX2 was knocked out in the T24T cells, the *HUR* mRNA levels were significantly higher than that in the T24T(Vector) cells (Figure 4B). The mRNA degradation experiments showed that there were similar *HUR* mRNA degradation rates between T24T(KO*SOX2*) and T24T(Vector) cells (Figure 4C), suggesting that HUR was regulated at the transcriptional level. Therefore, we constructed an HUR promoter-driven luciferase reporter and transfected it into T24T(Vector) and T24T(KO*SOX2* C1/C2) cells. As shown in Figure 4D, the HUR promoter transcriptional activity in T24T(KO*SOX2* C1/C2) cells was significantly higher than in T24T(Vector) cells, demonstrating that SOX2 regulated *HUR* mRNA transcription. To assess the mechanism by which the SOX2 transcriptional regulation of HUR, we evaluated transcription factors that may bind to the HUR promoter (Figure 4E). E2F1 and p-c-Jun were comparable in the T24T (Vector) and T24T (KOSOX2 C1/C2) cells, and C-Jun was decreased in T24T (KOSOX2 C1/C2) cells, while the Sp1 levels were increased in the T24T (KOSOX2 C1/C2) cells (Figure 4F). To assess the role of Sp1, the Sp1 shRNA constructs were transfected into T24T(KO*SOX2*) cells to establish stable transfectants T24T(KO*SOX2*/Nonsense), T24T(KO*SOX2*/sh*Sp1*#1), and T24T(KO*SOX2*/sh*Sp1*#2) cells (Figure 4G). Sp1 knockdown impaired HUR protein expression. Consistently, FOXO1 protein expression was also significantly attenuated in T24T(KO*SOX2*/sh*Sp1*#1 and #2) cells (Figure 4G). The above results revealed that SOX2 might exhibit an inhibitory effect on *HUR* transactivation by attenuating Sp1 expression, thereby downregulating FOXO1 expression. To test whether Sp1 was able to bind to the *HUR* promoter, a ChIP assay was carried out, and the region of the HUR promoter containing a tentative Sp1-binding site was found to be present in the immunoprecipitated complex of 293T(Anti GFP-Sp1), but not in 293T(IgG) (Figure 4H), strongly indicating that Sp1 indeed interacted with the HUR promoter. Moreover, Sp1 knockdown in T24T(KO*SOX2*) cells also dramatically increased the invasion ability (Figure 4I,J). These results demonstrated that Sp1 was a crucial transcription factor for HUR transcriptional activation and its downregulation of invasion abilities.

### 2.5. SOX2-Promoted SKP2 Transcription and in Turn Mediated Sp1 Protein Degradation

To elucidate the underlying mechanism of the SOX2 inhibition of Sp1 expression, the expression of *Sp1* mRNA was examined, and there was no significant change in the *Sp1* mRNA levels in T24T(KO*SOX2* C1/C2) and T24T(Vector) cells (Figure 5A), indicating that Sp1 might be regulated at levels of protein degradation or translation. To test this, MG132 and Cycloheximide (CHX) were employed. A degradation assay showed that the marked protein degradation of accumulated Sp1 could be observed within 6 h in T24T(Vector) upon CHX incubation, while this degradation was not seen for as long as 12 h in T24T(KO*SOX2*) cells (Figure 5B), demonstrating that SOX2 negatively regulated Sp1 protein stability. The ubiquitin-proteasome system is one of the major mechanisms responsible for protein degradation in eukaryotes [23]; Ubiquitin ligating enzyme E3, such as FBW7, SKP2, ITCH, XIAP, plays a vital role in the regulation of such process [24]. Heat shock proteins, HSP70 and HSP90, are the two most common types of chaperone proteins, which play a role in the degradation and regulation of misfolded peptides [25]. To evaluate which protein was involved in the SOX2 regulation of Sp1 degradation, proteins associated with protein degradation were examined in T24T(KO*SOX2* C1/C2) vs. T24T(Vector) cells. The expression of SKP2 was dramatically lower in T24T(KO*SOX2* C1/C2) cells than in T24T(Vector), while there were no observable changes of FBW7, SKP1, ITCH, XIAP, HSP70, and HSP90 proteins (Figure 5C). To test the role of SKP2 in the regulation of Sp1, SKP2 with Myc-tag was overexpressed in T24T(KO*SOX2*) cells. As shown in Figure 5D, Myc-SKP2 was successfully overexpressed in T24T(KO*SOX2*/Myc-SKP2). As expected, Sp1, HUR, and FOXO1 protein expression were markedly attenuated in T24T(KO*SOX2*/Myc-SKP2) compared to T24T(KO*SOX2*/Vector). Consistently, the protein degradation rates of Sp1 in T24T(KO*SOX2*/myc-SKP2) were greatly faster than that in T24T(KO*SOX2*/Vector) (Figure 5E). Further, the cell invasion ability of T24T(KO*SOX2*/myc-SKP2) was also higher than that of T24T(KO*SOX2*/Vector) (Figure 5F,G). These results indicated that SOX2 downregulates Sp1 expression through SKP2 promoting Sp1 protein degradation, which in turn led to the downregulation of HUR and FOXO1 expression, thereby improving BC cell invasion. To characterize the detailed mechanism by which SOX2 promoted SKP2 protein expression, we further examined The *sKP2* mRNA levels. The *SKP2* mRNA levels in T24T(KO*SOX2* C1/C2) cells were significantly lower than that in T24T(Vector) cells (Figure 5H), indicating that SOX2 regulated SKP2 at the mRNA level. The *SKP2* promoter activity in T24T(KO*SOX2* C1/C2) was also significantly lower than that in T24T(Vector), indicating that SOX2 regulated SKP2 at the mRNA transcriptional level (Figure 5I). SOX2 is a transcription factor that activates downstream gene transcription. To address whether SOX2 transcriptionally activated SKP2, we applied TFANSFAC^®^ Transcription Factor Binding Sites Software (geneXplain GmbH, Wolfenbüttel, Germany) for analyzing the SKP2 promoter. The SKP2 gene promoter region contains the putative DNA-binding sites for E2F1, AP1, SOX2, and Sp1 (Figure 5J), suggesting the possibility that SOX2 may bind to the SKP2 promoter region and directly activate SKP2 transcription. To test this, a ChIP assay was carried out. The SKP2 promoter sequence was found to be present in 293T cell extracts precipitated with Anti-SOX2 antibody, but not in 293T cell extracts pulldown with control IgG, strongly indicating that SOX2 indeed interacted with the SKP2 promoter (Figure 5K). These results revealed that SOX2 was a transcription factor for SKP2 transcriptional activation and its upregulation of BC invasion abilities.

### 2.6. SOX2 Promoted the Transcription of NCL, which Increased MMP2 mRNA Stability and Protein Expression, Thereby Promoting BC Cell Invasion

Given our previous studies demonstrated that NCL stabilizes *MMP2* mRNA and promotes the invasion of human BC T24T cells [26], we anticipated that SOX2 may promote MMP2 protein expression. This notion was substantially supported by the results shown in Figure 6A that *MMP2* mRNA levels were significantly downregulated in T24T(KO*SOX2* C1/C2) cells as compared with T24T(Vector) cells. The results from mRNA degradation experiments showed that *MMP2* mRNA degradation in T24T(KO*SOX2*) cells was accelerated compared with T24T(Vector) cells (Figure 6B), suggesting that SOX2 mediated MMP2 mRNA stabilization. Consistently, MMP2 protein expression was increased in T24T(KO*SOX2*/GFP-NCL) cells as compared with T24T(KO*SOX2*/Vector) (Figure 6C), supporting our notion that NCL is a downstream effector of SOX2 for the stabilization of MMP2 mRNA. This notion was approved by the results showing that the degradation rates of *MMP2* mRNA in T24T(KO*SOX2*/GFP-NCL) was reduced as compared with T24T(KO*SOX2*/Vector) cells (Figure 6D). Moreover, the cell invasion ability of T24T(KO*SOX2*/GFP-NCL) was also significantly increased compared with T24T(KO*SOX2*/Vector) cells (Figure 6E,F). The above results indicated that SOX2 could also promote BC cell invasion through the NCL stabilization of MMP2 mRNA in T24T cells. The effect of SOX2 on regulation NCL and MMP2 was also validated in U5637 cells (Appendix A). To elucidate the mechanism by which SOX2 regulated NCL, we first examined the mRNA expression changes of *NCL*. *NCL* mRNA expression was significantly lower in T24T(KO*SOX2* C1/C2) cells than in T24T(Vector) cells (Figure 6G), whereas there was no significant difference in the mRNA degradation rates between T24T(Vector) and T24T(KO*SOX2*) (Figure 6H). This suggested that SOX2 might regulate NCL expression at the transcriptional level. The potential transcription factors that are predicted to bind to the *NCL* promoter were analyzed and the results indicated that SOX2 might bind the putative DNA-binding site of the *NCL* gene promoter region (Figure 6I). To test whether SOX2 was able to bind to the *NCL* promoter, a ChIP assay was carried out as shown in Figure 6J. The *NCL* promoter sequence was found to be present in the immunocomplex that was pulled down with Anti-SOX2 but not with control IgG, strongly indicating that SOX2 indeed interacts with the *NCL* promoter. These results demonstrated that SOX2 was also critical for NCL transcriptional activation and its upregulation of invasion abilities. We also validate the SKP2-Sp1-HUR-FOXO1 and nucleolin-MMP2 axes in the TCGA database. The results indicated that SKP2 mRNA levels in bladder normal tissues were decreased as compared with bladder cancer tissues (Appendix A). Given our results showing that SKP2 regulated Sp1 at the level of protein degradation, there is no significant change in Sp1 mRNA in the TCGA database (Appendix A). We analyzed their association with the overall survival of BC patients and found that BC patients with high expressions of NCL and MMP2 exhibited a decreased overall survival than patients with low expressions of NCL and MMP2. On the other hand, BC patients with high expressions of HUR exhibited an increased overall survival than patients with low expressions of HUR (Appendix A). However, the gene expression correlation between SOX2 and other related gene expression, including SKP2, SP1, MMP2, HUR, NCL, and FOXO1, in the TCGA database did not show consistent correlated changes (Appendix A). This could be due to the heterogeneity of the many subtypes of human BCs, whereas the TCGA database lacks this kind of information. The IHC results also showed that HUR expression was decreased and SKP2 expression was elevated in N-butyl-N-(4-hydroxybutyl) nitrosamine (BBN)-induced mouse bladder cancers (Appendix A). Our results conclusively indicated that SOX2 promoted BC cell invasion through SKP2/Sp1/HUR/FOXO1 and NCL/MMP2 axes as summarized in Figure 6K.

## 3. Discussion

The transcription factor SOX2 is a member of the SRY-related HMG-box (SOX) family, which plays an essential role in determining the fate of cells, thereby regulating the development process of cells [27]. So far, previous research on SOX2 has emphasized its key role in stem cell maintenance, determining the lineage fate and essential factors for the pluripotency of somatic reprogramming. SOX2 has been shown to be upregulated and functionally relevant to various cancer, including bladder cancer [11,28,29,30,31,32,33,34]. SOX2 mediates a variety of cell functions, including the increased cell proliferation, invasion, migration, metastasis, and self-renewal of CSCs. To date, few studies have investigated the potential mechanisms underlying SOX2’s association with BC invasion. The present study demonstrates that SOX2 promotes BC invasion through two independent pathways, FOXO1 and MMP2, which provides a potential new direction for targeted therapy of BC in the future.

The FOXO family acts primarily as transcription factors and has a regulatory role in cell proliferation, apoptosis, ROS response, metabolism, cell cycle, longevity, and cancer biology [35,36,37]. FOXO1 regulates the expression of a number of cancer-related genes, such as p27^KIP1^, p21^WAF1^, cyclin D1, cyclin D2, FasL, p130, and Bim [38,39,40,41]. It plays critical biologic roles as a tumor suppressor in regulating cell-cycle arrest, apoptosis, DNA damage repair, invasion, and/or oxidative stress resistance [42,43]. Our previous studies show that miR-145 upregulates the anchorage-dependent growth of metastatic T24T BC cells by directly targeting the FOXO1 mRNA 3′-UTR [44]. We have also demonstrated that the anticancer drug ISO specifically inhibits BBN-induced invasive BC formation in vivo and human BC invasion in vitro by upregulating FOXO1 expression. The current research further showed that SOX2 promoted BC invasion by downregulating FOXO1 expression. Mechanistic research showed that SOX2 downregulated HUR, which is an RNA-binding protein that stabilizes target transcripts by blocking the degradation of mRNA and thus reduces the stability of FOXO1 mRNA. Further studies showed that HUR was regulated at the transcriptional level. RNA-IP experiments showed that the transcription factor Sp1 could bind to the promoter region of HUR, thereby promoting the transcription of HUR.

In the current study, we found that SOX2 activated SKP2 at the transcriptional level and promoted the degradation of the Sp1 protein. SKP2 is an F-box protein and E3 ubiquitin ligase that is involved in many key cellular processes such as cell cycle regulation, apoptosis, senescence, and the regulation of cancer stem cells [45,46]. The SKP1-CUL1-ROC1-F-box (SCF) complex is a multi-protein RING of an E3 ubiquitin ligase that degrades other proteins, including p27, p130, p57, Tob1, and c-myc [47]. SKP2 is an oncogene and is overexpressed in a variety of cancers [48,49], and associated with multiple tumor progressions and poor prognosis [46,50]. Our current study found that SKP2 could degrade Sp1 protein and promote the invasion of BC cells through the Sp1/HUR/FOXO1 signal axis. It is interesting to note that the promoter region of SKP2 contains an Sp1 binding site with which Sp1 can transcriptionally activate the expression of SKP2 [50]. In turn, SKP2 can promote the degradation of the Sp1 protein, suggesting that there may be a negative feedback mechanism to regulate Sp1 expression.

Previous studies have shown that SOX2 enhances the migration and invasion of laryngeal cancer cells and ovarian cancer cells through the upregulation of MMP2 expression [15,51]. SOX2 can inactivate FOXO1 in embryonic stem cells, while FOXO1 can inhibit SOX2 transcription in breast cancer cells [52,53]. Our current study found that SOX2 not only promoted BC cell invasion through the SKP2/Sp1/HUR/FOXO1 pathway but also promoted BC invasion by enhancing NCL transcription and upregulating MMP2. Given our previous studies have shown that ISO inhibits invasive BC formation in vivo and human BC invasion in vitro by targeting the STAT1/FOXO1 Axis, and MMP2 is negatively transcriptional regulated by FOXO1 and acts as a FOXO1 downstream effector for the ISO inhibition of BC invasion [16], we evaluated whether SOX2-initiated FOXO1 downregulation led to MMP2 induction. To test this, shRNA was used to knock down FOXO1 in T24T(KOSOX2) cells. The results indicated that FOXO1 knockdown did not show an observable effect on the MMP2 protein level (Figure 2G), suggesting that the SOX2-initiated FOXO1 was not involved in MMP2 regulation. Others also have shown that SKP2 also promotes cellular invasion and metastasis by inducing matrix metalloproteinases (MMPs) [54]. However, our results revealed that the knockdown of FOXO1 did not show observable effects on MMP2 expression T24T(KOSOX2) cells, whereas the knockdown of MMP2 does not affect FOXO1 in T24(SOX2) cells, suggesting that these two pathways are independent in SOX2-initiated human BC invasion.

## 4. Materials and Methods

### 4.1. Reagents, Antibodies, and Plasmids

A TRIzol reagent and a SuperScript™ First-Strand Synthesis system were purchased from Invitrogen (Grand Island, NY, USA). Actinomycin D (Act D) was purchased from Santa Cruz (Dallas, TX, USA). The specific antibodies against GAPDH were bought from Genetex (Irvine, CA, USA); the anti-XIAP antibody was purchased from BD (Franklin Lakes, NJ, USA); antibodies specifically against SOX2(23064S), FOXO1(2880S), GFP(2956S), Src(2109S), HSP70(4872S), SKP1(2156S), ITCH(12117S), Flag(2368S), p-c-Jun Ser63(2361S), p-c-Jun Ser73(3270S), c-Jun (9165S), and CREB(9197S) were purchased from Cell Signaling Technology (Beverly, MA, USA); the antibodies specific for HUR(sc-5261), RhoGDIα(sc-360), RhoGDIβ(sc-11359), Sp1(sc-14027), E2F1(sc-251), MMP-2(sc-10736), and SKP2(sc-7164) were obtained from Santa Cruz Biotechnology (Santa Cruz, CA, USA). The antibody against HSP90(SPA-830) was purchased from StressGen (San Diego, CA, USA). The antibodies against AUF1(ARP40238_T100) and FBW7(ARP47419_P050) were bought from Aviva Systems Biology (San Diego, CA, USA). The antibody against β-Actin(66009-1-Ig) was purchased from Proteintech (Rosemont, IL, USA). The shRNA specifically targeting human *HUR*, *FOXO1*, and *Sp1* were purchased from Open Biosystems (Pittsburg, PA, USA). The Clustered Regularly Interspaced Short Palindromic Repeats (CRISPR)/Cas9 system-specific targeting SOX2 was constructed into a pLenti-U6-sgRNA-SFFV-Cas9-2A-Puro Vector backbone which was inserted into the BbsI restriction site. The plasmid for the FLAG-tagged wild type (Flag-FOXO1) was described previously [55]. The SOX2(#16577), Myc- SKP2(#19947), and GFP- Sp1(#39325) expression plasmids were obtained from Addgene (Cambridge, MA, USA). The GFP-HUR expression plasmid was a generous gift from Dr. Imed-Eddine Gallouzi (McGill University Health Center, McGill University, Montreal, QC, Canada). GFP-nucleolin (GFP-NCL) expression plasmid was kindly provided by Dr. Michael B. Kastan (Comprehensive Cancer Center, St. Jude Children’s Research Hospital, Memphis, TN, USA) [56]. Human FOXO1 promoter luciferase reporter was kindly provided by Dr. Jean-Baptiste Demoulin (De Duve Institute, Catholic University of Louvain, BE-1200 Brussels, Belgium) [57]. The human HUR promoter (−1205 to +267), SKP2 promoter (−1395 to +61), and NCL promoter were cloned into the pGL3-Basic luciferase reporter.

### 4.2. N-butyl-N-(4-hydroxybutyl) Nitrosamine-Induced Mouse Bladder Cancer Model and Immunohistochemistry (IHC)

All animal testing procedures were approved by the Committee on Animal Resources at Wenzhou Medical University. For 3–4 weeks, C57BL/6 male mice (*n* = 15/group) drank tap water containing 0.05% BBN (TCI America, Portland, OR, USA) in opaque bottles for 20 weeks, while negative control mice drank plain tap water. The drinking water was prepared twice a week, and the consumption was recorded to estimate BBN intake. The mice were sacrificed after 20 weeks of the experiment, and their bladders were collected and preserved in paraffin for pathological analysis and immunohistochemistry staining (IHC) [58].

Immunohistochemistry staining (IHC) was used to detect SOX2 expression between BBN- induced invasive BC tissues and negative control bladder tissues. IHC was performed using a specific antibody against SOX2 (GeneTex, Irvine, CA, USA). IHC was carried out according to the scheme described in our previous study [16,59]. Immunostaining images were obtained using computer image analysis system (AxioVision Rel. 4.6, Carl Zeiss, Oberkochen, Germany). IHC-stained sections were evaluated at 400× magnification. At least five representative staining areas were analyzed for each slice, and the optical density was calculated based on the typically captured photographs.

### 4.3. Cell Culture and Transfection

T24 and T24T cells were generously provided by Dr. Dan Theodorescu (University of Colorado Comprehensive Cancer Center, Denver, CO, USA) as described in our previous publications [26] and cultured in DMEM:F12 = 1:1 with 5% FBS. The 293T cells were cultured in DMEM with 10% FBS. The cells were transfected with plasmid DNA using PolyJet™ DNA In Vitro Transfection Reagent (SignaGen Laboratories, Gaithersburg, MD, USA). Stable transfectants were selected with the corresponding antibiotics for 3–4 weeks, depending on the transfected different antibiotic resistance plasmids.

### 4.4. Dual-Luciferase Reporter Assay

The dual-luciferase assay kit was purchased from Promega (Madison, WI, USA). BC cells were co-transfected with either the FOXO1, HUR, or SKP2 promoter-luciferase reporter constructs, together with the Renilla luciferase vector pRL-TK. After stabilization, the cells were treated with passive lysis buffer according to the dual-luciferase assay manual and then measured with a luminometer (Lumat LB9507, Berthold Tech., Bad Wildbad, Germany). In each analysis, the firefly luciferase signals were normalized to the Renilla luciferase signals to eliminate the transfection efficiency differences as previously described [60].

### 4.5. Real-Time Quantitative Reverse Transcription PCR (RT-qPCR)

Total RNA from the cells was isolated by the TRIzol reagent. The total RNA (5μg) was then used for reverse transcription with oligo dT primer by SuperScript^TM^ First-Strand Synthesis system IV (Invitrogen Carlsbad, CA, USA). Specific primer pairs were designed to amplify human *FOXO1* (forward: 5′-AAC CTG GCA TTA CAG TTG GCC-3′, reverse: 5′-AAA TGC AGG AGG CAT GAC TAC GT-3′), *Sp1* (forward: 5′-GCT ATG CCA AAC CTA CTC CA-3′, reverse: 5′-TGA TCG TGA CTG CCT GAG A-3′), *SKP2* (forward: 5′-ATG TGA CTG GTC GGT TGC-3′, reverse: 5′-GGA GGG TGG ACA CTT CTA T-3′), *MMP2* (forward: 5′-CAA GTG GGA CAA GAA CCA GA-3′, reverse: 5′-CCA AAG TTG ATC ATG ATG TC-3′), *NCL* (forward: 5′-ACC TAA TGC CAG AAG CCA GCC A-3′, reverse: 5′-TTG CCC GAA CGG AGC CGT C-3′), and *GAPDH* (forward: 5′-AGA AGG CTG GGG CTC ATT TG-3′, reverse: 5′-AGG GGC CAT CCA CAG TCT TC-3′). The cycle threshold (CT) value was measured, and the relative mRNA expression was calculated based on the value of 2^−ΔΔCT^ as described in our published studies [61].

### 4.6. Western Blot

The entire cells were washed twice with ice-cold PBS and then extracted using cell lysis buffer (10 mM pH7.4 Tris-HCl, 1% SDS, 1mM Na_3_VO_4_, and proteasome inhibitor) on ice. The materials were heated at 100 °C for 10 min, and then all the nucleic acids were destroyed with ultrasonic waves. Protein concentrations were measured using NanoDrop 2000 (Thermo Scientific, Holtsville, NY, USA). The cell extracts were subjected to Western blot analysis with each antibody. The protein bands specifically binding to the primary antibodies were detected using alkaline phosphatase (AP)-conjugates secondary antibody and ECF (enhanced chemifluorescence) Western blot analysis system (Amersham Pharmacia Biotech, Piscataway, NJ, USA) as previously described [62]. The results shown were from at least three independent experiments.

### 4.7. mRNA Degradation Experiment

The quantitative number of cells was inoculated in a six-well plate and incubated in a medium containing 0.1% FBS for 12 h, followed by a medium containing 10% FBS for 8 h. The cells were then treated with a medium containing Actinomycin D and 10% FBS for a certain period of time. The mRNA degradation was analyzed by quantitative RT-PCR as described above.

### 4.8. Cell Invasion Assay

According to the manufacturer’s instructions, Invasion assays were measured using the BD Falcon (354480) kits. After incubation by transwell assay, the cells on both sides were fixed with 3.7% formalin for 2 min, washed twice with PBS, transferred to 100% methanol for 20 min, washed twice, and stained with Giemsa (1:20 diluted with PBS at room temperature) in the dark for 15 min. Finally, the cells were washed twice again with PBS, and the non-invaded cells were scraped four times with cotton swabs (PBS wetted). Images were taken with an Olympus DP71 (Olympus, Tokyo, Japan). and the number of cells was calculated by the software “Image J” (Image J, NIH, Bethesda, MD, USA). described in the previous paper [63].

### 4.9. RNA-IP Assay

RNA-IP assay was performed as described previously [64]. The transient transfection efficiency of 293T cells is very high and convenient for transient transfection, so we use 293T to carry out RNA-IP assay. In brief, twenty-four hours after the transfection, 293T cells were extracted using polysome lysis buffer. Anti-GFP agarose beads A/G (Vector Laboratories, Burlingame, CA, USA) were added to the supernatant and rotated overnight at 4 °C. The beads were washed three times and re-suspended, then incubated at 55 °C for 30 min, occasionally mixed. RNA was extracted by TRIzol, and the mRNA present in the immune complex was identified by RT-PCR.

### 4.10. Chromatin Immunoprecipitation Assay (ChIP)

ChIP was performed using an EZ-CHIP kit (Millipore, Darmstadt, Germany). (Millipore Technologies) according to the manufacturer’s instructions as described previously [65]. Briefly, 293T cells were fixed with 1% formaldehyde for 10 min at room temperature, transferred to lysis buffer, and sonicated to produce 200 to 400 bp chromatin DNA fragments. After centrifugation (13,000× *g*, 4 °C, 10 min), the 10-fold diluted supernatant was incubated with anti-GFP or SOX2 antibody or control rabbit IgG overnight at 4 °C. The immune complex was captured by protein G-agarose beads saturated with salmon sperm DNA and then eluted with elution buffer. The reverse cross-linking of the protein-DNA complex with free DNA was carried out by incubation overnight at 65 °C. DNA was extracted for PCR analysis. The PCR products were separated on 2% agarose gel with ethidium bromide, and the image was scanned with UV light.

### 4.11. Statistical Analysis

The student’s *t*-test was used to determine the significant differences between the treated and untreated groups. Results are expressed as mean ± SD from at least three independent experiments. *p* < 0.05 was considered to be a significant difference between the compared groups. All data were analyzed using GraphPad Software (La Jolla, CA, USA).

## 5. Conclusions

In conclusion, we found that SOX2 is an oncogene that promotes the invasion of BC cells through two target proteins, MMP2 and FOXO1. SOX2 upregulates MMP2 by promoting NCL transcription, while the SOX2 transcriptional activation of SKP2 is responsible for the downregulation of FOXO1. Moreover, SKP2 promotes the degradation of Sp1 and reduces its transcriptional activation of HUR, which downregulates FOXO1 mRNA stability, and the downregulation of the tumor suppressor gene FOXO1 ultimately leads to the increased invasion of BC cells. In addition, the upregulation of MMP2 and the downregulation of FOXO1 are responsible for the SOX2 promotion of BC invasion. These findings not only provide new insights into the underlying mechanisms of BC invasion but also identify SOX2 as a new drug target for the inhibition of invasive BC formation, which can be used to prevent the metastasis of MIBC.

## Figures and Tables

**Figure 1 ijms-23-12532-f001:**
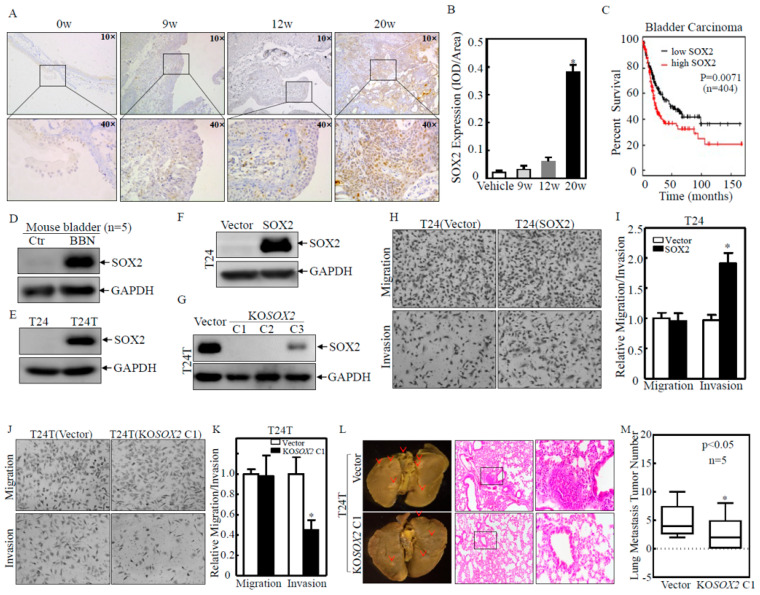
SOX2 was upregulated in mouse and human BC tissues and was critical for BC cell invasion in vitro and nude mouse lung metastasis in vivo. (**A**,**B**) Mouse bladder tissues (*n* = 15) with BBN treatment for a different time were collected for IHC staining to evaluate SOX2 protein expression. The representative images of each tissue were captured as described in the “Materials and Methods” section. SOX2 protein expression levels were analyzed by calculating the integrated IOD/area using Image-Pro Plus version 6.0 (Media Cybernetics, Rockville, USA). The results for each group were expressed as mean ± SD. Symbol (*) indicates a significant difference between the vehicle control group and the BBN-treated 20 weeks group (*p* < 0.05). (**C**) Kaplan–Meier estimation of SOX2 levels related to overall survival (OS) in BC patients from the TCGA Database. (**D**,**E**) Western blot was used to assess the expression of SOX2 in mouse bladder tissues with BBN treatment (Combined equal aliquots of bladder tissue from five mice) (**D**), human BC cells (**E**). (**F**,**G**) Western blot was used to evaluate SOX2 expression in the stable transfectants of SOX2 overexpression (**F**) or SOX2 knockout (**G**) in BC cells. (**H**–**K**) The invasion abilities were evaluated in SOX2 stable overexpression transfectants T24(SOX2) or SOX2 stable knockout transfectants T24T(KO*SOX2*, clone1), as well as their corresponding vector control transfectants, by using BD BioCoat^TM^ Matrigel^TM^ Invasion Chamber (BD, Bedford, MA, USA). The asterisk (*) indicates a significant difference in invasion ability in comparison to their corresponding vector control transfectants (*p* < 0.05). The bars are presented as the mean ± SD from three independent experiments. (**L**,**M**) T24T(KO*SOX2* C1), and its vector control cells were injected into the tail veins of BALB/c nude mice (2 × 10^6^ cells; five mice per group). After 6 weeks, the mice were sacrificed, and the lungs were removed and fixed in picric acid and photographed. The number of lung metastatic tumors was counted. The results are expressed as the mean ± SD. Student’s *t*-test was used to determine the *p*-value, and the asterisk (*) indicates a significant decrease (*p* < 0.05). The images of mouse lungs were captured using ZEISS SteREO Discovery.V20, and HE staining was used to confirm the metastatic colonization. Red arrow and black rectangle represent tumor mass and cells respectively.

**Figure 2 ijms-23-12532-f002:**
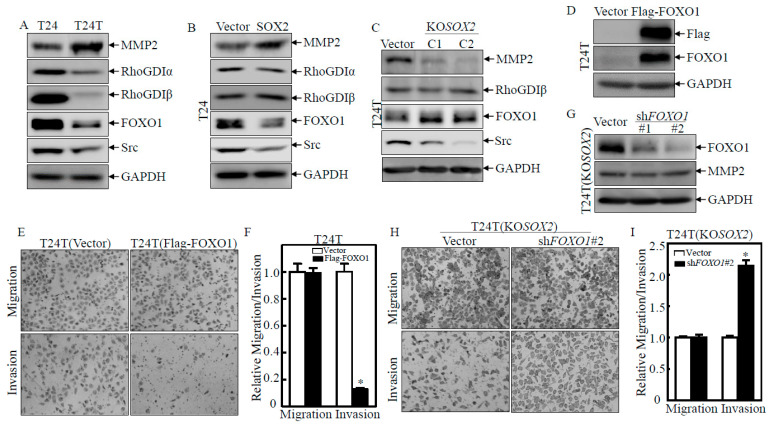
Upregulation of MMP2 and downregulation of FOXO1 play a crucial role in the invasive ability of BC cells. (**A**–**D**,**G**) The indicated cells were seeded into six-well plates. The cells were extracted upon the cell density reaching 80–90%, and the cell extracts were subjected to Western blot for the determination of protein expression as indicated. GAPDH or β-Actin was used as a protein-loading control. (**E**,**F**,**H**,**I**) The invasion abilities of the T24T(Vector) versus T24T(Flag-FOXO1) cells and T24T(KO*SOX2*/Vector) versus T24T(KO*SOX2*/sh*FOXO1*#2) were determined using a BD BioCoat^TM^ Matrigel^TM^ Invasion Chamber. The asterisk (*) indicates a significant difference in invasion abilities in comparison to their corresponding vector control transfectants (*p* < 0.05). The bars are presented as the mean ± SD from three independent experiments.

**Figure 3 ijms-23-12532-f003:**
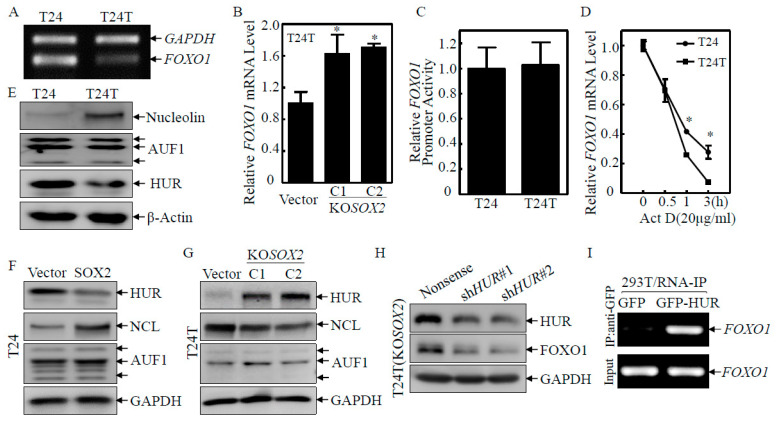
HUR downregulation was responsible for SOX2-mediated FOXO1 mRNA destabilization. (**A**,**B**) The indicated cells were extracted with TRIzol reagent to isolate total RNA upon the density reaching 80–90%. The *FOXO1* mRNA levels were determined with RT-PCR (**A**) or Real-time PCR (**B**) by using the specific primers. *GAPDH* was used as an internal control. The asterisk (*) indicates a significant difference in *FOXO1* mRNA level in comparison to their corresponding vector control transfectants (*p* < 0.05). (**C**) The indicated cells were transfected with a FOXO1 promoter-driven luciferase reporter together with pRL-TK. The transfectants were seeded into 96-well plates and then subjected to determine the FOXO1 promoter activity by measuring the luciferase activity. pRL-TK was used as an internal control to normalize the transfection efficiency. Each bar indicates the mean ± SD from three replicate assays. (**D**) T24 and T24T cells were seeded into six-well plates. After synchronization, the T24 and T24T cells were treated with actinomycin D (Act D) for the indicated time points, and then the total RNA was isolated and subjected to Real-time PCR analysis to evaluate the mRNA levels of *FOXO1* normalized to internal control *GAPDH*. The asterisk (*) indicates a significant difference in *FOXO1* mRNA level in comparison to T24T cell line (*p* < 0.05). (**E**–**H**) T24 and T24T stable transfectants as indicated were extracted, and the cell extracts were subjected to Western blot to determine the expression of the indicated proteins. GAPDH was used as a protein loading control. (**I**) 293T cells transfected with GFP-tagged HUR were used to perform an RNA-IP assay using an anti-GFP antibody to examine the interaction of HUR with *FOXO1* mRNA.

**Figure 4 ijms-23-12532-f004:**
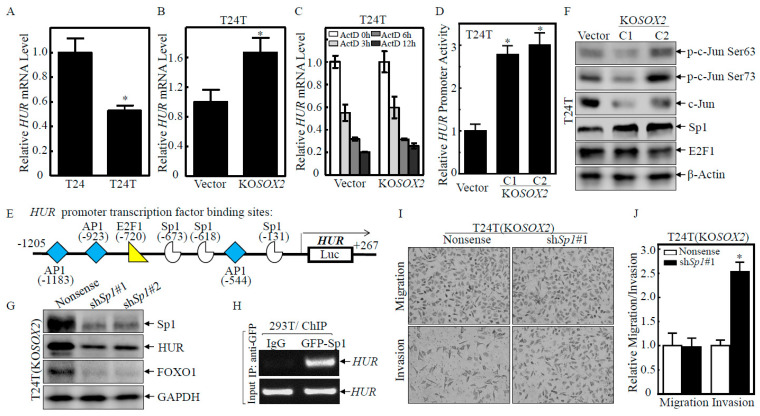
SOX2 inhibited HUR transcription by downregulating Sp1 protein expression. (**A**,**B**) The indicated cells were extracted with TRIzol reagent to isolate total RNA upon the density reaching 80–90%. *HUR* mRNA levels were determined with Real-time PCR by using the specific primers. *GAPDH* was used as an internal control. (**C**) T24T(Vector) and T24T(KO*SOX2*) cells were seeded into six-well plates. After synchronization, cells were treated with Act D for the indicated time points, then total RNA was isolated and subjected to Real-time PCR analysis to evaluate the mRNA levels of *HUR* and *GAPDH*. (**D**) The indicated cells were transfected with HUR promoter-driven luciferase reporter together with pRL-TK. The transfectants were seeded into 96-well plates and then subjected to determine HUR promoter activity. pRL-TK was used as an internal control to normalize transfection efficiency. Each bar indicates the mean ± SD from three replicate assays. (**E**) The diagram shows the transcription factors which could potentially bind to the human HUR promoter region. (**F**,**G**) Western blot was used to detect the protein levels of p-c-Jun Ser63, p-c-Jun Ser73, c-Jun, and Sp1 in T24T(Vector) and T24T(KO*SOX2* C1/C2) cells. β-Actin was used as a protein loading control. (**G**) shRNAs targeting the human Sp1 gene were transfected into T24T(KO*SOX2*) cells to knock down Sp1 expression stably. Western blot was used to analyze the indicated protein expression; GAPDH was used as the protein loading control. (**H**) The 293T cells with overexpressing GFP-tagged Sp1 were employed for ChIP assay using an anti-GFP antibody to test the interaction of Sp1 with *HUR* mRNA. (**I**,**J**) The invasion abilities of T24(KO*SOX2*/Nonsense) versus T24(KO*SOX2*/sh*Sp1*#1) cells were determined using a BD BioCoat^TM^ Matrigel^TM^ Invasion Chamber. The asterisk (*) indicates a significant difference in invasion abilities in comparison to its nonsense control transfectant (*p* < 0.05). The bars are presented as the mean ± SD from three independent experiments.

**Figure 5 ijms-23-12532-f005:**
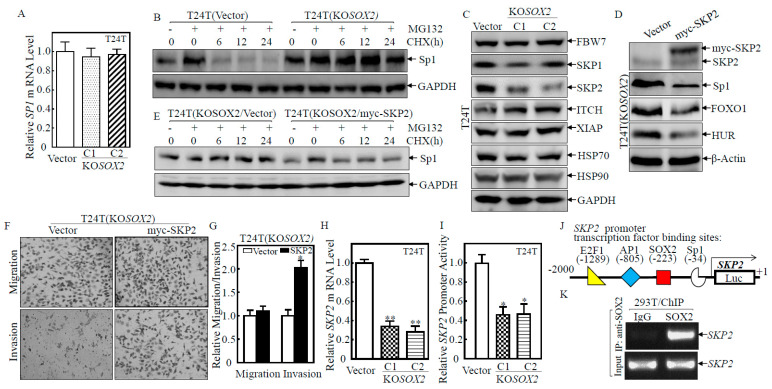
SKP2 mediated the invasion of BC cells by promoting the degradation of the Sp1 protein. (**A**) Real-time PCR was applied to compare the *Sp1* mRNA levels in T24T(Vector) and T24T(KO*SOX2* C1/C2) cells. (**B**,**E**). The indicated cells were pretreated with MG132 and the cells were then subjected to protein degradation assay in the presence of CHX for the indicated time points. Western blot was used to analyze Sp1 protein degradation rates between the indicated cells. β-Actin or GAPDH was used as a protein loading control. (**C**) The indicated stable transfectants were subjected to Western blot to determine SKP1, SKP2, HSP90, HSP 70, and ITCH protein levels. (**D**) The indicated stable transfectants were subjected to Western blot to determine Sp1, FOXO1, and HUR protein levels. (**F**,**G**) The invasion abilities of T24T(KO*SOX2*/Vector) and T24T(KO*SOX2*/myc-SKP2) cells were determined using a BD BioCoat^TM^ Matrigel^TM^ Invasion Chamber. The asterisk (*) indicates a significant difference in invasion abilities in comparison to its vector control transfectant (*p* < 0.05). The bars are presented as the mean ± SD from three independent experiments. (**H**) Real-time PCR was applied to compare the mRNA levels of SKP2 in T24T(Vector) and T24T(KO*SOX2* C1/C2) cells. The asterisk (**) indicates a significant difference in *SKP2* mRNA in comparison to its vector control transfectant (*p* < 0.01). (**I**) The indicated cells were transfected with SKP2 promoter-driven luciferase reporter together with pRL-TK. The transfectants were seeded into 96-well plates and then subjected to determine *SKP2* promoter luciferase activity. pRL-TK was used as an internal control to normalize transfection efficiency. Each bar indicates the mean ± SD from three replicate assays. (**J**) The diagram shows the transcription factors that could potentially bind to the human HUR promoter region. (**K**) The 293T cells with overexpressing SOX2 were tested by ChIP assay using an anti-SOX2 antibody to test the interaction of SOX2 with *SKP2* mRNA.

**Figure 6 ijms-23-12532-f006:**
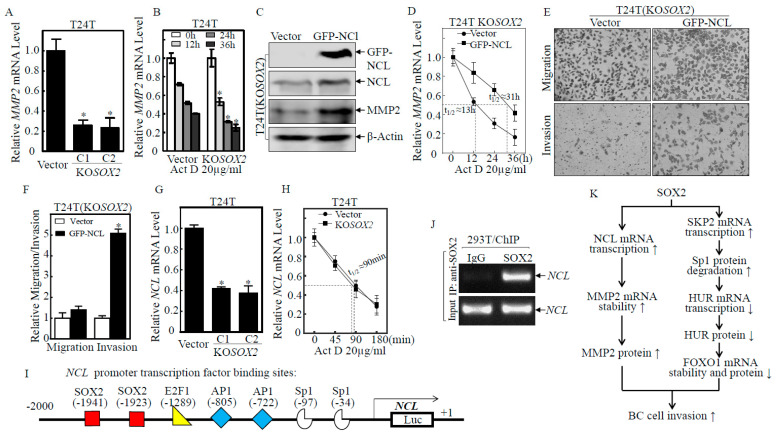
SOX2 promoted transcriptional nucleolin upregulation and in turn increased the stability of MMP2 mRNA and protein expression, further promoting the invasion of BC cells. (**A**,**G**) Real-time PCR was applied to compare the mRNA levels of *MMP2* and *NCL* in T24T(Vector) and T24T(KO*SOX2* C1/C2) cells. The asterisk (*) indicates a significant difference in *MMP2/NCL* mRNA in comparison to its vector control transfectant (*p* < 0.05). (**B**,**D**,**H**) The indicated cells were seeded into six-well plates. After synchronization, the cells were treated with Act D for the indicated time points, then total RNA was isolated and subjected to RT-PCR analysis to evaluate the mRNA levels of *MMP2* or *NCL*. (**C**) T24T(KO*SOX2*/Vector) and T24T(KO*SOX2*/GFP-NCL) cells were cultured in six-well plates until the cell density reached 80–90%. The cells were then extracted, and the cell extracts were subjected to Western blot to determine the expression level of the indicated proteins. β-Actin was used as a protein loading control. (**E**,**F**) The invasion abilities of T24T(KO*SOX2*/Vector) and T24T(KO*SOX2*/GFP-NCL) cells were determined using a BD BioCoat^TM^ Matrigel^TM^ Invasion Chamber. The asterisk (*) indicates a significant difference in invasion abilities in comparison to its vector control transfectant (*p* < 0.05). The bars are presented as the mean ± SD from three independent experiments. (**I**) Potential transcriptional factor binding sites in the *NCL* promoter region (−2000–+1) were analyzed using the TRANSFAC 8.3 engine online. (**J**) The 293T cells with overexpressed SOX2 were employed for ChIP assay using an anti-SOX2 antibody to test the interaction of SOX2 with *NCL* mRNA. (**K**) Schematic summary of molecular mechanisms underlying SOX2 promotion of invasion of human BC cells. SOX2 stabilizes MMP2 mRNA and thereby upregulates MMP2 protein by promoting transcription of NCL. In addition, SOX2 transcriptionally activates SKP2, which promotes the degradation of Sp1 and reduces its transcriptional activation of HUR, thereby downregulating FOXO1 mRNA stability and FOXO1 protein. Increased MMP2 and decreased FOXO1 together promote BC invasion.

## Data Availability

The datasets used and/or analyzed during the current study are available from the corresponding author on reasonable request.

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
