# Peer review of "SOX2 Promotes Invasion in Human Bladder Cancers through MMP2 Upregulation and FOXO1 Downregulation"

_ijms, 2022, doi:10.3390/ijms232012532_

Round 1

Reviewer 1 Report

This is a very interesting subject of research. The results obtained by the authors of the manuscript are consistent with modern ideas in the field of oncogenesis of bladder cancer and are worthy of publication.

There are minor questions:

1) about the specificity of the antibodies used in that study, so for some proteins - several zones are marked as related to one protein. Please, provide data that the antibodies (FOXO1, AUF1) are specific and an explanation of this phenomenon. Without this, there are doubts about the conclusions of the article;

2) the choice of cell lines for immune co-deposition and RNA studies is not sufficiently justified in the text of the manuscript.

Author Response

We appreciate the reviewer #1 for the overall evaluation: "This is a very interesting subject of research. The results obtained by the authors of the manuscript are consistent with modern ideas in the field of oncogenesis of bladder cancer and are worthy of publication. "

Comment 1: about the specificity of the antibodies used in that study, so for some proteins - several zones are marked as related to one protein. Please, provide data that the antibodies (FOXO1, AUF1) are specific and an explanation of this phenomenon. Without this, there are doubts about the conclusions of the article;

Response: We thank the reviewer for this suggestion. According to the FOXO1 (#2880S) datasheet [https://en.cellsignal.jp/products/primary-antibodies/foxo1-c29h4-rabbit-mab/2880], FOXO1 has a molecular weight of 78-82 kDa and frequently exhibits 2 bands. According to AUF1 (ARP40238_T100) product specification [https://www.avivasysbio.com/sd/tds/html_datasheet.php?sku=ARP40238_T100], AUF1 has a molecular weight of 37-45 kDa and frequently exhibits 3 bands. We believe that the WB results of FOXO1 and AUF1 are specific.

Comment 2: the choice of cell lines for immune co-deposition and RNA studies is not sufficiently justified in the text of the manuscript.

Response: We thank the reviewer for this suggestion. 293T cells are the most frequently used molecular biology tool cells because they are very efficiently transfected for immune co-deposition and RNA studies, etc [Zha Z, Han X, Smith MD, Liu Y, Giguère PM, Kopanja D, Raychaudhuri P, Siderovski DP, Guan KL, Lei QY, Xiong Y. A Non-Canonical Function of Gβ as a Subunit of E3 Ligase in Targeting GRK2 Ubiquitylation. Mol Cell. 2015 Jun 4;58(5):794-803. doi: 10.1016/j.molcel.2015.04.017]. We have discussed in the revised manuscript (Page 4).

Reviewer 2 Report

The study by Xie et al defines the mechanism by which SOX2 promotes invasion of human bladder cancer. The authors showed that SOX2 promotes BC invasion through two independent pathways. One route is through the down-regulation of FOXO1 expression via Sp1/HUR/FOXO1 pathway. The other route is via up-regulation of MMP2 in an NCL-dependent mechanism. The study is well carried out. The manuscript is well written, with well-described methods and figure legends and great attention to details. The manuscript is easy to follow. The manuscript will benefit from addressing the following concerns prior to publication.

1. Combining all aliquots into a single lane is not a general practice and could generate misleading conclusions. The individual replicates of the western blots should be run in individual lanes and the western blot data should be quantified.

2. Line 200:  can't conclude invasion ability by just probing for SOX2 levels in bladder tissue. This statement is better suited after describing Fig 1E.

3. Line 362: c-jun levels show a dip in the T24T KOSOX2 clones compared to the control in Fig 4F as opposed to what the authors claim in line 362. Please clarify.

4. Line 548: In the discussion, the authors cited global roles of FOXO1 in cell proliferation, cell cycle, and cancer biology. As the manuscript focuses on the invasive properties of bladder cancer cells, the authors should also cite literature where FOXO1 has been shown to specifically inhibit invasion abilities of cancer cells.

5. Line 501: typo, SKP2 mRNA levels in bladder normal tissues look decreased.

6. Fig 1: The y-axis in the migration/invasion quantification figures, for example, Fig 1i is misleading. As written, it appears the values are the ratio of migration to invasion whereas they are means of cell counts normalized to the control means. The y-axis label should be changed to relative cell count and the figure legend should mention that the means are normalized to vector control.

7. Fig. 2A: The authors should provide the rationale of why RhoGDI and Src proteins were investigated in this context. The levels of these proteins are different in the T24 and T24T cells, so the authors should explain why there were not investigated further.

Author Response

We appreciate the reviewer for the positive evaluation: "The study is well carried out. The manuscript is well written, with well-described methods and figure legends and great attention to details. The manuscript is easy to follow."

Comment 1: Combining all aliquots into a single lane is not a general practice and could generate misleading conclusions. The individual replicates of the western blots should be run in individual lanes and the western blot data should be quantified.

Response: Due to very limited amount of mouse bladder tissue from each mouse (half of each bladder tissue was used for RNA extraction and other half was used for protein extraction), we have to combine equal aliquots of bladder tissue from each mouse together (n=5) and then were subjected to Western blot for analysis of SOX2 protein expression.

Comment 2: Line 200:  can't conclude invasion ability by just probing for SOX2 levels in bladder tissue. This statement is better suited after describing Fig 1E.

Response: We thank the reviewer for this suggestion and related changes have been made in Line 205 of the revised manuscript.

Comment 3: Line 362: c-jun levels show a dip in the T24T KOSOX2 clones compared to the control in Fig 4F as opposed to what the authors claim in line 362. Please clarify.

Response: We thank the reviewer for this suggestion and we have made changes in the revised manuscript.

Comment 4: Line 548: In the discussion, the authors cited global roles of FOXO1 in cell proliferation, cell cycle, and cancer biology. As the manuscript focuses on the invasive properties of bladder cancer cells, the authors should also cite literature where FOXO1 has been shown to specifically inhibit invasion abilities of cancer cells.

Response: We thank the reviewer for this suggestion and we have added reference in the revised manuscript.

Comment 5: Line 501: typo, SKP2 mRNA levels in bladder normal tissues look decreased.

Response: We are grateful to the reviewer for her/his carefulness and the related corrections were made in the revised version.

Comment 6: Fig 1: The y-axis in the migration/invasion quantification figures, for example, Fig 1i is misleading. As written, it appears the values are the ratio of migration to invasion whereas they are means of cell counts normalized to the control means. The y-axis label should be changed to relative cell count and the figure legend should mention that the means are normalized to vector control.

Response: We thank the reviewer for this suggestion. Relative migration was normalized to the corresponding control, and relative invasion activity was calculated after re-normalization to cell migration. So, the y-axis represents the relative migration and relative invasion.

Comment 7: Fig. 2A: The authors should provide the rationale of why RhoGDI and Src proteins were investigated in this context. The levels of these proteins are different in the T24 and T24T cells, so the authors should explain why there were not investigated further.

Response: We thank the reviewer for this suggestion and we have explained in the revised manuscript (Page 7).

Reviewer 3 Report

Qipeng Xie et al. reported the SOX2 function in the BC invasion with its mechanism by SKP2-Sp1-HUR-FOXO1 and nucleolin-MMP2 dual axes. The study was conducted scientifically and well-designed. I would suggest a minor revision before publication.

Please address the below concerns:

1.       Line 22 and 23, HUR transcription / FOXO1 mRNA mentioned as gene, should be italic.

2.       Article title: if using the “upregulation” in the whole manuscript, please be consistent in the title as well, and delete the dash -.  Including Line 253, line 377.

3.       Line 34, I don’t think the word “variants” is appropriate here.

4.       Please insert a citation for BBN induced mouse model in Line 89-95. And method 2.2 title text format is not right. And animal model is performed in NYU School of Medicine? Should put some affiliation in the acknowledgment?

5.       Figure 1-M, suggest changing Y axis title to Lung metastasis nodules number.

6.       Line 247, 6 should be superscript, above the line of text.

7.       Line 219 and 246 SOX2 should be italic.

8.       Line 407 CHX needs full name.

9.       Figure in manuscript is not high-resolution but great in original-images file. Make sure they will publish and use high-resolution ones.

10.    Figure 1, considering you showed the RhoGDI and Src of WB in the figure, please give a sentence of description in the result text.

11.    If necessary, could add the mRNA degradation assay method part.

12.    Please add a few discussions of SOX2 with MMP2 and FOXO1 based on other papers in other cancer types maybe. 

Author Response

We appreciate the reviewer for the overall evaluation: "Qipeng Xie et al. reported the SOX2 function in the BC invasion with its mechanism by SKP2-Sp1-HUR-FOXO1 and nucleolin-MMP2 dual axes. The study was conducted scientifically and well-designed. I would suggest a minor revision before publication."

Comment 1: Line 22 and 23, HUR transcription / FOXO1 mRNA mentioned as gene, should be italic.

Response: We are grateful to the reviewer for her/his carefulness and the related corrections were made in the revised version.

Comment 2: Article title: if using the “upregulation” in the whole manuscript, please be consistent in the title as well, and delete the dash -.  Including Line 253, line 377.

Response: We are grateful to the reviewer for her/his carefulness and the related corrections were made in the revised version.

Comment 3: Line 34, I don’t think the word “variants” is appropriate here.

Response: We thank the reviewer for this suggestion and we have made changes in the revised manuscript.

Comment 4: Please insert a citation for BBN induced mouse model in Line 89-95. And method 2.2 title text format is not right. And animal model is performed in NYU School of Medicine? Should put some affiliation in the acknowledgment?

Response: We thank the reviewer for this suggestion and we have added reference in the revised manuscript. And method 2.2 The title text format has been modified. The animal experiment was conducted at the Laboratory Animal Center of Wenzhou Medical University which was approved by the Ethical Committee of Wenzhou Medical University.

Comment 5: Figure 1-M, suggest changing Y axis title to Lung metastasis nodules number.

Response: We thank the reviewer for this suggestion and we have made changes in the revised manuscript.

Comment 6: Line 247, 6 should be superscript, above the line of text.

Response: We are grateful to the reviewer for her/his carefulness and the related corrections were made in the revised version.

Comment 7: Line 219 and 246 SOX2 should be italic.

Response: We are grateful to the reviewer for her/his carefulness and the related corrections were made in the revised version.

Comment 8: Line 407 CHX needs full name.

Response: We are grateful to the reviewer for her/his carefulness and the related corrections were made in the revised version (Page 11).

Comment 9: Figure in manuscript is not high-resolution but great in original-images file. Make sure they will publish and use high-resolution ones.

Response: We thank the reviewer for this suggestion. Our images are in high definition and may have lost clarity due to the format becoming PDF during upload. We will upload high-definition images later.

Comment 10: Figure 1, considering you showed the RhoGDI and Src of WB in the figure, please give a sentence of description in the result text.

Response: We thank the reviewer for this suggestion and we have made modifications in the revised manuscript (Page 7).

Comment 11: If necessary, could add the mRNA degradation assay method part.

Response: We thank the reviewer for this suggestion and we have added mRNA degradation assay in method part in the revised manuscript (Page 3).

Comment 12: Please add a few discussions of SOX2 with MMP2 and FOXO1 based on other papers in other cancer types maybe.

Response: We thank the reviewer for this suggestion and we have added a few discussions of SOX2 with MMP2 and FOXO1 in the revised manuscript (Page 15).